## RESEARCH ARTICLE

# Brief psychological intervention for the prevention of deliberate self-poisoning: A randomized controlled trial from Sri Lanka

Lakmini De Silva[1]*, Judi Kidger[2], Sampath Tennakoon[3], Andrew Dawson[1,4], Indika Gawarammana[1,5], Thilini Rajapakse[1,6]

1 South Asian Clinical Toxicology Research Collaboration, Faculty of Medicine, University of Peradeniya, Peradeniya, Sri Lanka, 2 Bristol Medical School, Population Health Sciences, University of Bristol, Bristol, United Kingdom, 3 Department of Community Medicine, University of Peradeniya, Peradeniya, Sri Lanka, 4 Royal Prince Alfred Hospital, Camperdown, New South Wales, Australia, 5 Department of Medicine, Faculty of Medicine, University of Peradeniya, Peradeniya, Sri Lanka, 6 Department of Psychiatry, Faculty of Medicine, University of Peradeniya, Peradeniya, Sri Lanka

* lakminidesilva@yahoo.com

## Abstract

Previous work suggests that many nurses in Sri Lanka, particularly those who work in primary care, involved in the medical treatment of persons who attempt self-poisoning already approach their role holistically and consider 'providing mental health support as a part of nursing. However, at present, nurses are not given formal training on the delivery of such support, nor has the efficacy or feasibility of such an intervention been assessed in Sri Lanka. A mixed-method design was employed: a pilot randomized controlled trial with embedded qualitative methods. Participants (n = 300) were admitted to Teaching Hospital Peradeniya after non-fatal self-poisoning incidents. The brief psychological intervention consisted of a counselling session that encouraged participants to explore alternative strategies for managing emotional distress and future stressors. Ward nurses received training to deliver the intervention to assigned patients. Quantitative and qualitative data were collected at baseline and follow-up. At the six-month follow-up, the trail demonstrated the intervention's effectiveness in significantly reducing anxiety and promoting positive coping strategies, although no significant differences were observed in depression rates or alcohol use disorder. Qualitative interviews indicated that participants, found the counselling valuable for emotional support and guidance in managing future interpersonal conflicts and stressors. Overall, this study suggests that brief psychological interventions can support the mental health of those at risk of self-poisoning and encouraging alternative coping strategies. When offering mental health support to individuals who have attempted self-poisoning, it is more feasible to assign dedicated, trained nurses than to expect all nurses to contribute.

**Data availability statement:** We do not have explicit consent from study participants to make the data publicly available. Participants were informed at the outset that every effort would be taken to protect their privacy and confidentiality, and all data were de-identified. Due to these ethical restrictions, the minimal dataset underlying the findings of this study cannot be shared publicly. Data access requests may be submitted to the South Asian Clinical Toxicology Research Collaboration (SACTRC), Faculty of Medicine, University of Peradeniya, at enquiry@sactrc.org, which serves as a non-author institutional point of contact. Requests will be reviewed in accordance with institutional governance policies and the conditions of the original ethical approvals. The data are securely stored and managed by SACTRC at the Faculty of Medicine, University of Peradeniya, and will be retained in accordance with institutional data retention policies to ensure long-term availability.

**Funding:** This work was supported by the National Research Council, Sri Lanka (Grant No. NRC-16-052 to TR) and the Translational Australian Clinical Toxicology (TACT) Program, Pilot Grant (to LDS), awarded as an encouragement for early-career researchers. The funders had no role in study design, data collection and analysis, decision to publish, or preparation of the manuscript.

**Competing interests:** The authors have declared that no competing interests exist.

## Introduction

Deliberate self-harm (DSH) in the form of attempted or deliberate self-poisoning (DSP) continues to be a serious public health issue in Sri Lanka. DSH is the strongest predictor of suicide, after suicidal ideation [1]. Bans of low-toxicity pesticides were associated with a drop in pesticide case fatality, but were not associated with reductions in self-poisonings, and overall hospitalizations remained high [2]. There has been a shift toward less lethal means, especially medicinal overdose [2], and an increase in the number of adolescents and young people engaging in deliberate self-harm in Sri Lanka [3,4].

In Sri Lanka, the most common method of DSH is attempted or deliberate self-poisoning [5,6]. Interpersonal conflict with close others and financial stress are frequently reported as key triggering factors associated with self-harm, in LMIC countries, particularly in South Asian countries such as India and Sri Lanka [5,7,8]. In particular, being female, the experience of childhood adversity, domestic violence, poverty and alcohol use increases vulnerability to deliberate self-harm in young people [9,10].

Brief psychological interventions (BPI) are a form of short psychological intervention that are effective in reducing self-harm and suicide attempts [8,11]. Given that this can be administered with minimal training and is not time consuming, these brief interventions have greater potential to be integrated into routine clinical practice without the need for significant additional resources or extensive reconfiguration of existing services. Therefore, it is worthwhile to explore the efficacy of brief psychological interventions (BPI) in Sri Lanka, considering the resource limitations, especially in mental health care.

In Sri Lanka, DSH patients are typically managed in general medical and surgical wards, involving a combination of medical and psychiatric care; a proportion of patients are evaluated by overburdened psychiatrists through ward referral practices [12,13]. Hence, brief psychological intervention delivered by nurses would be feasible and efficacious in reducing suicidal ideation and improving coping skills among the Sri Lankan population. Thus the overall aims of this study were to develop: 1) an evidence-based counseling intervention, aimed at reducing the risk of future self-harm by improving the ability to cope with acute emotional distress, which can be delivered by nurses after training, and to explore the feasibility and acceptability of such an interventional strategy; and 2) to explore whether such a intervention produces meaningful outcomes in reducing suicidal ideation and repetition of self-harm, improving distress tolerance coping skills and rates of depression and anxiety, at one-year follow-up.

## Materials and methods

### Study design

This study employed a mixed-methods approach [14]; a randomised controlled trial (RCT) with an embedded qualitative study.

## Setting

Teaching Hospital Peradeniya is a large tertiary care teaching hospital in the Central Province of Sri Lanka. Patients from the Central Province who have attempted self-poisoning and who require medical care are admitted to the Toxicology Unit (Ward 17) of this hospital.

## Participants

Individuals invited for this study were all patients admitted to the study hospital for medical management after an episode of DSP. Both first attempters and repeat attempters of DSP aged between 16–64 years, and who did not require inpatient psychiatric treatment, were included in the study. Individuals who reported or carried documents indicating a previous diagnosis of intellectual disability or dementia, who were too physically unwell to participate in the interview prior to discharge from the hospital, and those who could not speak Sinhala were excluded from the study.

## Procedure

Block randomisation was used to allocate participants randomly to either the brief psychological intervention (BPI) or the control group- treatment as usual [15,16]. Randomisation was performed utilising blocks of four, ensuring equal allocation to intervention and control groups, using a pre-generated list of random numbers, produced by using specific software.

Qualitative research has the potential to explore perspectives not easily assessable within an RCT design [17]. Qualitative methods were used in parallel to the quantitative methods employed in the pilot RCT. After the patient gave written informed consent, the participants were randomly allocated to either the BPI arm (intervention group) or the treatment as usual arm (control group). The assessor conducting the six-month and one-year assessments in the pilot RCT was blinded to the case/control status of the participants. There was no blinding during the qualitative interviews.

The study period was over a consecutive four-month period (from November 1, 2017, to March 29, 2018); the endpoint of the study was 12 months after the baseline assessment, and this was completed as planned without ever stopping. Participants were assessed at three different time points: baseline (T1), six months (T2) and 12 months (T3) post-intervention. The full trial protocol has been published previously [18].

## Trial intervention

**The brief psychological intervention (BPI).** The intervention consisted of a structured nurse-delivered BPI focused on supporting the individual to learn to tolerate distress and cope in more adaptive ways. This was derived partly from one section of a previous intervention for self-harm carried out in Pakistan – the C-MAP intervention [8]. The original C-MAP intervention was a brief, problem-focused therapy conducted over six sessions within three months and consisted of multiple components, including an evaluation of the self-harm attempt, crisis skills, problem-solving, CBT techniques to manage emotions, negative thinking, interpersonal relationships, and relapse prevention strategies [8]. We examined the distress tolerance component of the C-MAP intervention and used those concepts to develop our intervention. The BPI in this study focused more on helping the participant learn or plan alternative ways to deal with future distress (rather than self-harming) – such as using basic techniques to manage negative emotions (for example, strategies such as emotional ventilation, relaxation, distraction or self-soothing strategies) [8,19]. A special focus was placed on encouraging the participant herself/himself to come up with strategies that would suit them to deal with distress; the nurse's role was to listen to the person and to guide the discussion to encourage the person to discuss alternative ways of dealing with distress in future. Given that our intervention was limited to one session with the participant, the other components of the C-MAP study intervention were not included in our intervention. The BPI was focused on identifying how the individual could react to acute distress in more adaptive ways in future (Fig 1).

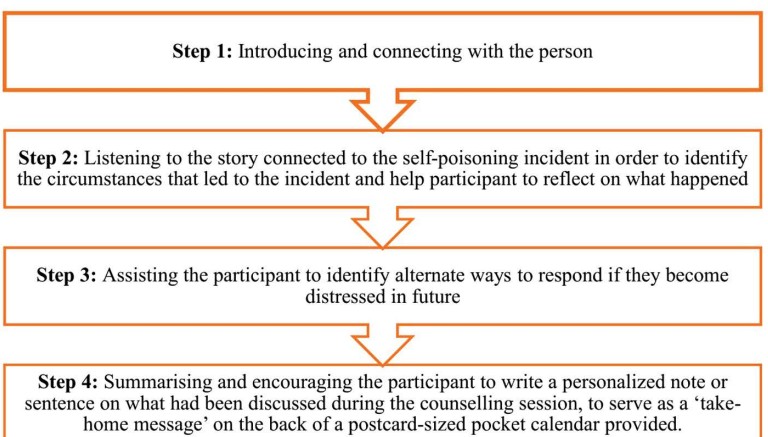

**Fig 1. Steps and content of the brief psychological intervention.** This figure illustrates the key stages that was followed by the nurses in delivering the intervention, including introducing, listening to the story, assisting participants to identify alternate coping strategies, summarising and writing a take-home message.

**Training of nurses in the delivery of the BPI.** All nurses working in the Toxicology Unit of the Teaching Hospital Peradeniya at the time of starting the study were given a brief overview of the project by the research team and were invited to participate in the BPI training. This training was conducted in two workshops, by a consultant psychiatrist (TR) with experience in this field. Participants engage in role-play with varying case scenarios. The nurses were trained in how to encourage the participants to write notes and to keep a pocket calendar with the message as a reminder of alternate ways of coping with distress. A brief manual that outlined the steps of the counselling intervention in simple terms was also provided to the nurses to be used as a guide when delivering the intervention.

Even though all the nursing staff in the Toxicology Unit (ward 17) participated in the training workshop with enthusiasm and showed interest in contributing towards the study, after the study started, it became apparent that due to other ward commitments and time constraints, the nurses found it difficult to contribute to delivering BPI. This problem worsened during the dengue epidemic, when many additional medical patients were also admitted to the Toxicology Unit (ward 17). Considering this situation, and after discussion with the nurses of the Toxicology Unit (ward 17) and all the investigators of the study, it was decided to amend the project so that the BPI was to be delivered by a designated, trained nursing officer. The intervention, the study participants, the follow-up and all other details of the study remained unchanged.

**Treatment as usual.** Both the intervention and control groups received medical care and TAU during their hospital stay. This included medical management as well as psychiatry referral, as deemed required by the medical team (which is the usual practice), irrespective of the intervention/control status of the participant.

## Sample size

In this study, we considered changes in the severity of depression as one of the primary outcome measures because it is an important predictor of deliberate self-harm [20]. Previous interventional studies have examined changes in the severity of depression using the Patient Health Questionnaire PHQ-9 as an outcome measure [21]. One study reported that patients in the intervention group had significantly fewer depressive symptoms compared with patients in the usual care group at the 12-week follow-up, and the PHQ-9 mean score at the three-month follow-up was 2.4 and 7.1 in the intervention and control group, respectively (4.7 mean difference) [21]. The difference between the groups was significant (p < .001) using baseline scores as the covariate. Using an independent sample t-test with a two-sided significance level of 0.05, a sample size of 15 in each group would give 80% power to detect a difference in the means of 4.7, assuming

that the estimated common standard deviation is 4.49 [21]. Based on similar calculations and allowing for a 50% drop-out rate, we estimated that a minimum of 30 participants in each group would be required for this study; however, since this study was conducted among those who had attempted self-poisoning, we anticipated (based on previous studies) that a little less than 50% of the sample would have depression [22]. This study aimed to examine clinically relevant changes in depression levels (PHQ-9 scores above 10, indicating a moderate level of depression or above, rather than in the < 10 range quoted in the previous study). To allow for this greater margin, we inflated the sample size to 150 cases and 150 controls in the RCT.

The sampling strategy was a purposive one for the qualitative study, aiming to recruit participants who could provide information-rich accounts [23], and the guiding principle of selecting sample size was data saturation [24,25]. Participants were selected to ensure variation in age, gender, employment, ethnicity and rural/ urban origin. The participants were recruited for the qualitative study on three occasions: soon after delivery of the intervention and during the hospital stay for the management of the self-harm attempt (T1), six months (T2) and one year (T3) following the incident.

### Primary outcomes

**Depression.** The primary outcome was depression at the T2 and T3 follow-ups as measured by the Patient Health Questionnaire-9 (PHQ-9) [26,27], which has been translated into Sinhala and validated for use in Sri Lanka [26].

**Anxiety.** Anxiety was assessed with the GAD-7 [28]. There were no locally validated versions of this scale available at the time of the study. The scale was translated into Sinhala and back-translated and checked for accuracy before use.

**Alcohol use disorders.** The Alcohol Use Disorders Identification Test (AUDIT) [29] was used to measure change in alcohol use disorders. The current study used the adapted and locally validated Sri Lankan version of the AUDIT [29].

**Coping skills.** The intermediate outcome was the assessment of coping skills as measured by the Brief Coping Inventory (Brief COPE) [30]. The Brief COPE was used to assess patients' coping skills. At the time of the study, no locally validated tools for assessing coping were available. The scale was translated to Sinhala and back-translated and checked for accuracy prior to use.

### Secondary outcomes

**Level of suicide intent and rates of repetition of self-harm.** The Pierce Suicide Intent Scale (PSIS) [31] was used to assess the level of suicidal intent. The scale has been translated into Sinhala, validated, and used previously in Sri Lanka [32].

### Qualitative in-depth interviews

Qualitative interviews were conducted with the study participants in both the BPI and TAU groups. The aim of the interviews was to explore the efficacy, feasibility and acceptability of the counselling intervention (Table 1).

T1 interviews were conducted in the Toxicology Unit (Ward 17), where the participants were being treated (after medical recovery), and T2 and T3 in a private room at the research facility. Prior to all the interviews, the researcher informed or reminded all of the participants about the study in simple terms, and they were invited to ask questions or raise any concerns about the study. Two separate interview guides were used for the baseline assessment and the follow-ups. (Table 1). The guides were tested through pilot interviews to ensure the relevance and clarity of the questions before the main data collection began. The researcher recorded important notes in a field notebook after conducting the interviews.

The interviews were approximately 45 minutes to one hour in duration and were conducted confidentially, in person, in Sinhala and carried out until data saturation occurred. All interviews were audio-taped and lasted approximately 45–60 minutes. All interviews were transcribed for future analysis.

**Table 1. Details of in-depth interviews conducted at three different time points.**

| Time Point | Group | Focus Areas |
| --- | --- | --- |
| T1 | Intervention group | Identify triggers with the DSP event<br>Exploring coping methods<br>Experience of the intervention |
| | Control group | Identify triggers with the DSP event<br>Exploring coping methods |
| T2 and T3 | Intervention group | Exploring coping methods<br>Recurrent suicidal ideation<br>Experience of the intervention |
| | Control group | Exploring coping methods<br>Recurrent suicidal ideas |

T1- baseline assessment, T2- six-month follow-up assessment, T3- twelve-month follow-up assessment.

## Data analysis

**Quantitative data analysis.** The quantitative data were entered into and analysed using the statistical package for the social sciences (SPSS 20). A strict intention-to-treat approach was followed in the analysis. The study and control groups were described using univariate analysis (frequencies, means) and age comparisons were made using Student's t-test. In addition to categorical analyses, continuous PHQ-9 scores were analyzed using t-tests to examine differences between groups, consistent with the sample size calculation and original protocol. Continuous analyses comparing mean PHQ-9, GAD-7, and AUDIT scores between groups were also conducted using t-tests, consistent with the sample size calculation and original protocol. For AUDIT, continuous analyses included both male and female participants; however, for categorical AUDIT analyses, only male participants were included, as none of the female participants reported alcohol use at any assessment point. Levels of depression, as assessed by the PHQ-9, were collapsed into two categories (not depressed vs. depressed) (Kroenke et al., 2001) to maintain adequate numbers in each category, given the limited number of participants at follow-up. The categories of drinking generated by the AUDIT were collapsed into two categories, where the low-risk category remained as it was, but the hazardous drinking, harmful use and probable dependence categories were considered as a single category, named the high-risk category. The levels of anxiety in the GAD-7 were also re-categorised as low level of anxiety vs. higher level of anxiety for the purpose of the analysis. This categorisation also aligned with the pre-specified primary categorical outcomes for this pilot trial. While we acknowledge that categorising continuous scales may reduce statistical power and the ability to detect differences between groups, and may underestimate effect sizes where differences are observed, the decision to use categorical endpoints was consistent with the pre-specified primary outcomes of this feasibility trial and was necessary to ensure adequate group sizes given the high attrition at follow-up.. Chi-square tests were used to explore the association between the BPI vs. TAU and categorical outcomes of depression (PHQ-9), anxiety (GAD-7), and alcohol use (AUDIT) at six-month and one-year follow-up. Outcomes that were collapsed into binary categories (e.g., PHQ-9, GAD-7, AUDIT) were analyzed using chi-square tests to explore group associations, and Relative Risk was calculated for these categorical outcomes as specified in the protocol. This approach allowed effect estimation consistent with the protocol while accommodating the small sample size of this pilot feasibility trial. Statistical significance was assessed at the 95% confidence level. Planned multiple imputation, as specified in the original protocol if missing data exceeded 20%, was not conducted because the attrition pattern was non-random (e.g., younger participants were disproportionately lost) and follow-up data were missing in entire blocks for many individuals, and analyses were therefore restricted to complete cases.

This study was designed as a pilot feasibility trial, and the analysis plan focused on simple group comparisons rather than modelling changes across timepoints. The primary aim was to assess feasibility and acceptability, with effect

estimation being exploratory. The use of more sophisticated methods, such as repeated-measures analysis, is acknowledged as appropriate for a full-scale trial.

**Method of qualitative data analysis.** This study applied methodological triangulation of both quantitative and qualitative data in order to enhance the complexity of the data set and confidence in the analysis. The interviews were analysed using thematic analysis methods, a method for identifying, analysing, and reporting patterns (themes) within data [33]. Once a research interview had been transcribed and checked, the interview was read and re-read several times, notes were made in relation to reflections on the interview and its context, and then coding began. The analysis of each interview began with initial (line-by-line) coding followed by a process of identifying themes within the interviews. The researcher identified the themes by considering patterned responses or meanings within the dataset. This led to some themes being combined to generate overarching themes. The themes were discussed regularly with two other team members during development to test their coherence and distinctiveness from each other.

### Ethical consideration

Ethical approval for this study was obtained from the Ethical Review Committee of the Faculty of Medicine, University of Peradeniya (2016/EC/81), and administrative approval was obtained from the hospital authorities as required. The trial was registered in the Sri Lanka Clinical Trials Registry SLCTR/2017/008 on 21.03.2017. The trial was designed according to the Standard Protocol Items: Recommendations for Interventional Trials (SPIRIT) 2013 checklist and reported in accordance with the CONSORT 2010 Statement: updated guidelines for reporting parallel group randomized trials [34]. The study also followed consolidated criteria for reporting qualitative research (COREQ) [35].

Every effort was made to ensure that there were no interruptions or adverse effects on each participant's usual treatment (cases and controls). Participants were referred to further psychiatric care and follow-up where required.

All data were kept strictly confidential, and all participants were informed that they were participating in a research study and that data (anonymously) may be presented in research reports and publications.

### Researcher reflexivity

LDS is a qualitative researcher with a PhD and a background in sociology, who led the interviews and analysis. Her prior exposure to individuals who have self-harmed occurred during her MPhil work, which shaped her interest in and approach to the topic, as well as the data collection and analysis. She has worked in the field of suicide and self-harm research in Sri Lanka for over 10 years.

### Results

A total of 300 patients who had self-harmed were screened for inclusion in the study, and 293 completed the baseline assessment and were randomized (Fig 2).

A total of 149 patients were randomized to the BPI group and 144 to the TAU group (Table 2). Of the 293 participants, 185 (63.1%) were women, and the mean age was 27.2 years (S.D. = 12.6); 88 (59%) were single.

At the baseline assessment (T1), the participants in interviewed consisted of twelve women and four men, while at the six-month follow-up (T2), 15 women and nine men participated, and at the one-year follow-up (T3), twenty women and ten men were interviewed (Table 3).

We carried out 16 in-depth interviews during the baseline assessment (T1), 24 in-depth interviews at the six-month follow-up (T2) and 30 in-depth interviews at the one-year follow-up (T3). Three participants participated in the interviews conducted at the all-time points while six participants participated in both the T1 and T2 phases and T2 and T3 phases, respectively.

Five participants refused consent at T1, while two and three participants refused consent to participate in the interview at baseline T2 and T3, respectively. The quotes given in the results have been presented by mentioning the participants'

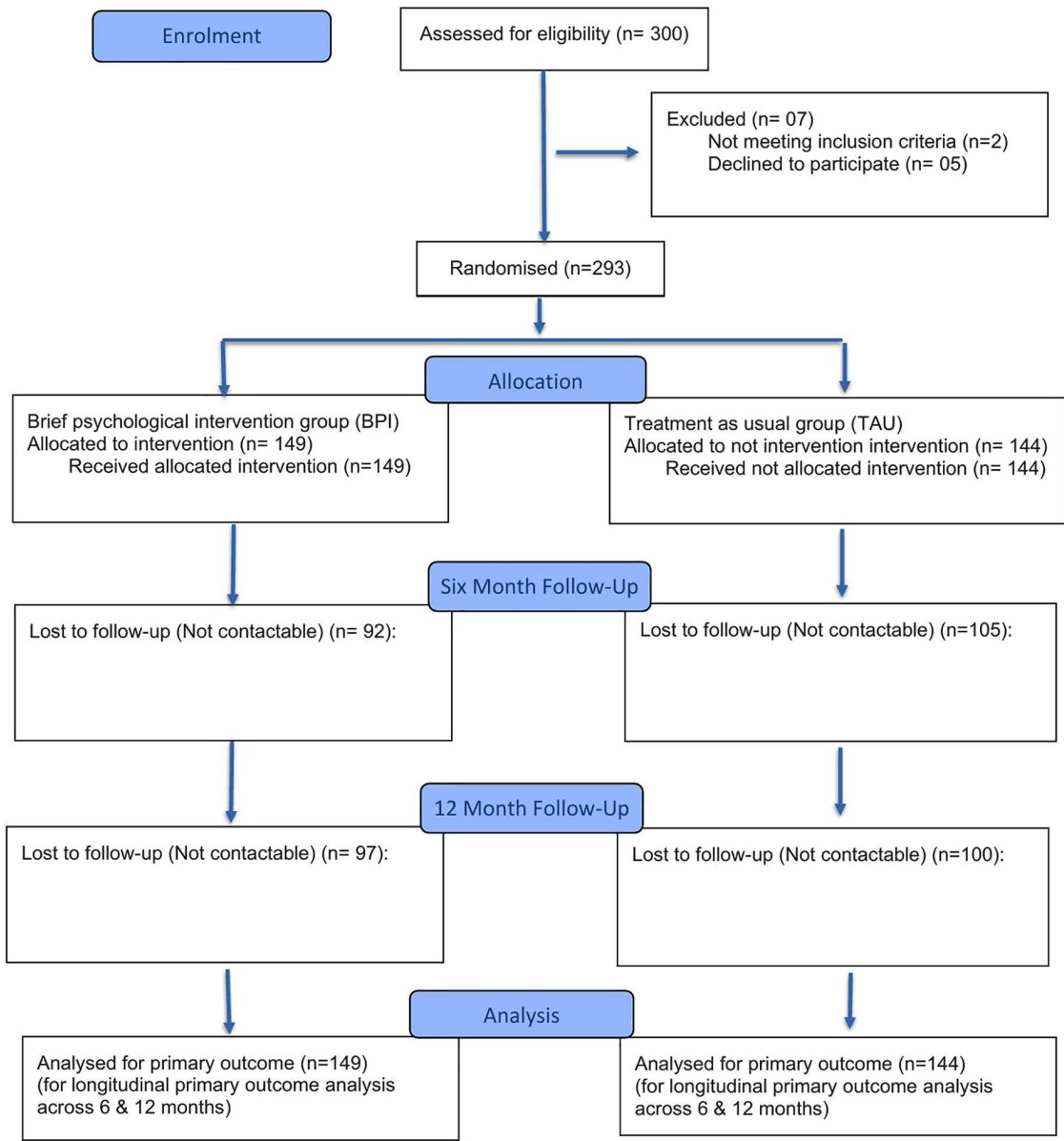

**Fig 2. Flow chart of participant recruitment.** This figure illustrates the flow of participant recruitment throughout the study, including the number of individuals assessed for eligibility, excluded, enrolled and follow-ups conducted.

ID and (P ID) and based on the time point: baseline assessment (T1), six-month follow-up (T2) and twelve-month follow-up (T3).

## Analysis of the clinical symptoms

At the 6-month and 12-month follow-ups, no differences were found between the TAU group and BPI group in the prevalence of depression or alcohol use disorder (Table 4). With regards to the rates of anxiety, there was a significant difference $\chi^2$ (1, N = 89) = 3.66, p = 0.05) at the six-month follow-up between the intervention and control

**Table 2. Socio-demographic characteristics of participants by treatment group.**

|  | BPI(n = 149) | TAU (n = 144) | P |
|---|---|---|---|
|  | n (%) | n (%) |  |
| Age, years: mean (SD) | 26.1 (13.5) | 28.0 (11.6) | 0.19 (t -1.29) |
| Female, n (%) | 99 (66.4) | 86 (59.7) | 0.26 (χ² 1.24) |
| Marital status, n (%) |  |  | 0.01 (χ² 11.1) |
| Single | 88 (59) | 58 (40.2) |  |
| Married | 61 (40.9) | 86(59.7) |  |
| Ethnic group, n (%) |  |  |  |
| Sinhala | 130(87.2) | 119(82.6) | 0.49 (χ² 1.39) |
| Tamil | 13(8.7) | 18(12.5) |  |
| Moor | 6 (4) | 7 (4.8) |  |

BPI -Brief - Counselling Intervention group, TAU- Treatment as Usual group.

**Table 3. Socio-demographic characteristics of interview participants.**

|  | T1 | T2 | T3 |
|---|---|---|---|
| **Age*** |  |  |  |
| Mean | 25.9 | 29.7 | 28.2 |
| Range | 17-60 | 16-64 | 16-64 |
| **Gender, n (%)*** |  |  |  |
| Female | 12 (75%) | 15(62.5%) | 20(66.7%) |
| Male | 4 (25%) | 9 (37.5%) | 10(33.3%) |
| **Marital Status, n (%)** |  |  |  |
| Married | 7 (44%) | 14 58.3% | 16(53.3%) |
| Single | 9 (56%) | 10(41.7%) | 14(46.7%) |
| **Educational Level, n (%)*** |  |  |  |
| Having/ had Grade 01–10 Education | – | 4 (16.7%) | 6 (20%) |
| Up to Ordinary Level examination (O/L) | 10(63%) | 13(54.2%) | 11(36.6%) |
| Up to Advanced level (A/L) | 4 (25%) | 7 (29.1%) | 12 (40%) |
| Engaged in/ completed Tertiary Education | 2 (13%) | – | 1 (3.4%) |
| **Employment, n (%)*** |  |  |  |
| Employed | 4 (25%) | 5 (20.8%) | 7 (23.3%) |
| Self-employed | – | 2 (8.3%) | 2 (6.7%) |
| Unemployed | 7(43.7%) | 9 (37.5%) | 7 (23.3%) |
| Student | 5(31.3%) | 8 (33.4%) | 14(46.7%) |

*At the time of the interview.

two groups, with the intervention group having a lower level of anxiety (Table 4). Nevertheless, this difference could not be seen at the one-year follow-up χ²(1, N = 101) = 0.13, p = 0.71. Results from the continuous analyses generally supported the patterns observed in the categorical analyses, though exact trends differed slightly (Table 5).

At the 6-month and 12-month follow-ups, no differences were found between the TAU group and BPI group in rates of repeat attempts and suicidal thoughts (Table 6).

**Table 4. Comparison of Anxiety, Depression, and Alcohol Use Disorder (AUDIT) at the baseline and follow-ups.**

| | BPI Group n (%) | TAU group n (%) | chi-square value | Relative Risk (RR) | P |
|---|---|---|---|---|---|
| **Depression (PHQ-9)** | | | | | |
| **Baseline assessment (n=293)** | **n=149** | **n=144** | | | |
| Not depressed | 83(55.7) | 85 (59.0) | 0.33 | 1.08 | 0.56 |
| Depressed (screened positive) | 66 (44.3) | 59 (41.0) | | | |
| **Six-month follow-up (n=89)** | **n=50** | **n=39** | | | |
| Not depressed | 35 (70) | 25 (64.1) | 0.34 | 0.84 | 0.55 |
| Depressed (screened positive) | 15 (30) | 14 (35.9) | | | |
| **One-year follow-up (n=100)** | **n=56** | **n=44** | | | |
| Not depressed | 34(75.4) | 32 (72.7) | 1.58 | 1.44 | 0.2 |
| Depressed (screened positive) | 22(24.6) | 12 (27.3) | | | |
| **Alcohol use disorder (AUDIT)** | | | | | |
| **Baseline assessment (n=108)** | **N=50** | **N=58** | | | |
| Low risk for alcohol use | 31 (62) | 32 (55.2) | 0.51 | 0.85 | 0.47 |
| Higher risk for alcohol use* | 19 (38) | 26 (44.8) | | | |
| **Six-month follow-up (n=27)** | **N=15** | **N=12** | | | |
| Low risk for alcohol use | 10(66.7) | 06 (50) | 0.76 | 0.67 | 0.38 |
| Higher risk for alcohol use* | 05(33.3) | 06 (50) | | | |
| **One-year follow-up (n=30)** | **N=16** | **N=14** | | | |
| Low risk for alcohol use | 11 (68.8) | 10 (71.4) | 0.02 | 1.09 | 0.87 |
| Higher risk for alcohol use* | 5 (31.2) | 04 (28.6) | | | |
| **Anxiety (GAD-7)** | | | | | |
| **Baseline assessment (n=293)** | **n=149** | **n=144** | | | |
| Low levels of anxiety | 92 (61.) | 89 (61.8) | 1.13 | 1.00 | 0.99 |
| Higher levels of anxiety | 57 (38.3) | 55 (38.2) | | | |
| **Six-month follow-up (n=89)** | **n=50** | **n=39** | | | |
| Low levels of anxiety | 43(88.4) | 27(71.8) | 3.66 | 0.46 | 0.05 |
| Higher levels of anxiety | 07(11.6) | 12(28.2) | | | |
| **One-year follow-up (n=101)** | **n=57** | **n=44** | | | |
| Low levels of anxiety | 47(82.5) | 35(81.8) | 0.13 | 0.86 | 0.71 |
| Higher levels of anxiety | 10(17.5) | 09(18.2 | | | |

BPI -Brief Counselling Intervention, TAU- Treatment as Usual.

Anxiety (GAD-7)- Low levels of anxiety- minimal or mild anxiety, Higher levels of anxiety- moderate or severe anxiety.

*Six-month follow-up non-respondent=n=2 (BPI group).

Depression (PHQ-9- Not depressed -minimal or mild depressive symptoms. Depressed (screened positive)- moderate, or moderately severe, or severe depressive symptoms.

* Six-month follow-up -non-respondent=02 (BPI group),01 (TAU group).

Alcohol use disorder (AUDIT).

*Hazardous Drinking or Probable Harmful Use, or Probable Dependence group.

## Changes in coping skills

At the baseline assessment (before intervention delivery), there were no significant differences between the intervention and TAU groups in the use of coping strategies, except for behavioural disengagement (p=0.02), acceptance (p=0.02) and self-blame (p=0.04), which were significantly more common among the participants in the intervention group (Table 7).

**Table 5. Mean score for Anxiety, Depression, and Alcohol Use Disorder (AUDIT) at the baseline and follow-ups.**

|  | BPI Group | TAU | Mean Difference | P |
|---|---|---|---|---|
|  | Mean (S.D.) n | Mean (S.D.) n |  |  |
| Patient Health Questionaire-9 (PHQ-9) |  |  |  |  |
| Baseline | 9.52(7.26)149 | 8.97(7.13)144 | -0.552 | 0.51 |
| Six-month follow-up | 6.32(5.66)50 | 7.77(7.46)39 | 1.449 | 0.3 |
| Twelve-month follow-up | 8.23(6.38)56 | 7.30(7.42)44 | -0.937 | 0.5 |
| Alcohol Use Disorders Identification Test (AUDIT) |  |  |  |  |
| Baseline |  |  |  |  |
| Six-month follow-up | 1.99(4.14)149 | 3.02(6.09)144 | 1.034 | 0.09 |
| Twelve months follow-up | 1.40(3.77)50 | 2.77(6.28)39 | 1.369 | 0.2 |
|  | 1.82(4.59)57 | 2.09(5.32)44 | 0.266 | 0.78 |
| Generalized Anxiety Disorder 7 (GAD-7) |  |  |  |  |
| Baseline | 7.78(6.00)149 | .49(6.48)144 | -0.292 | 0.68 |
| Six-month follow-up | 4.56(4.92) 50 | 6.28(6.59)39 | 1.722 | 0.16 |
| Twelve-month follow-up | 5.00 (4.75)57 | 5.27(6.23)44 | 0.273 | 0.8 |

BPI -Brief Counselling Intervention, TAU- Treatment as Usual; (SD) (standard deviation).

**Table 6. Number of repeat episodes of self-harm and suicidal ideation, and repeat attempts at six-month and 12-month follow-ups.**

|  | Six months | | | | 12 months | | | |
|---|---|---|---|---|---|---|---|---|
| Outcome | BPI group | TAU | RR | P | BPI group | TAU group | RR |  |
|  | (n=52) | group |  |  | (n=57) | (n=44) |  | p |
|  | n (%) | (n=39) |  |  | n (%) | n (%) |  |  |
|  |  | n (%) |  |  |  |  |  |  |
| **Any suicidal ideation** |  |  |  |  |  |  |  |  |
| Yes | 14(26.9) | 13(33.3) | 0.81 | 0.5 | 18(31.5) | 15(34.0) | 0.92 |  |
| No | 38 (73) | 26(66.6) |  |  | 38(66.6) | 28(63.6) |  | 0.77 |
| Non-respondents | 0(0) | 0(0) |  |  | 1(1.7) | 1(2.2) |  |  |
| **Any self-harm attempts** |  |  |  |  |  |  |  |  |
| Yes | 4(7.6) | 4(10.2) | 0.75 | 0.66 | 6(10.9) | 6(13.6) | 0.78 |  |
| No | 48(92.3) | 35(89.7) |  |  | 49(89) | 37(84.0) |  | 0.64 |
| Non-respondents | 0(0) | 0(0) |  |  | 2 | 1(2.2) |  |  |

RR-Relative Risks.

BPI -Brief Counselling Intervention group, TAU- Treatment as Usual group.

Non-respondents at one-year follow-up: n=2 (BPI group), 1 (TAU group).

At the six-month follow-up, participants in the intervention group were more likely to report using coping strategies such as self-distraction (p=0.02) and instrumental support (p=0.03), as compared to the control group. The use of other coping strategies did not differ significantly between the two groups. At the one-year follow-up assessment, there were no statistically significant differences between the intervention and TAU groups, except for one strategy: participants in the control group were significantly more likely to use acceptance as a coping method (p=0.04) compared to the intervention group.

### Facing life's challenges: ways of coping following the self-harm attempt

Using qualitative interviews, we were able to gain a more in-depth insight into the coping strategies used by the participants at baseline and the six- and 12-month follow-ups. Some had positively managed their later life problems with the

**Table 7. Mean scores for coping at baseline, six months and one year as a function of treatment group.**

| Brief COPE | Baseline | | | | | | | Six-month follow-up | | | | | | | One-year follow-up | | | | | | |
|---|---|---|---|---|---|---|---|---|---|---|---|---|---|---|---|---|---|---|---|---|---|
| | BPI | | TAU | | t | df | p | BPI | | TAU | | t | Df | P | BPI | | TAU | | t | df | P |
| | N | Mean (SD) | N | Mean (SD) | | | | N | Mean (SD) | N | Mean (SD) | | | | N | Mean (SD) | N | Mean (SD) | | | |
| Self-distraction | 144 | .5 (1.9) | 42 | 5.4 (1.9) | 0.39 | 284 | 0.69 | 50 | 6.1 (1.3) | 39 | 5.4 (1.8) | 2.32 | 87 | 0.02 | 57 | 6.0 (01.6 | 44 | 6.3 (1.8) | -0.85 | 99 | 0.39 |
| Active coping | 145 | 5.6 (1.8) | 139 | 5.4(2.0) | 0.65 | 282 | 0.51 | 49 | 6.3 (1.6) | 39 | 6 (1.8) | 0.78 | 86 | 0.43 | 57 | 6.1 (1.7) | 44 | 6.1 (1.7) | -0.10 | 9 | 0.91 |
| Denial | 143 | 4.2 (1.9) | 138 | 3.9 (1.9) | 1.46 | 279 | 0.14 | 49 | 4.4 (1.7) | 39 | 4.7 (1.9) | -0.57 | 86 | 0.56 | 57 | 4.4 (1.9) | 44 | 5.0 (2.0) | -1.55 | 9 | 0.12 |
| Substance use | 142 | 2.5 (1.6) | 140 | 2.6 (1.5) | -.49 | 280 | 0.62 | 50 | 2.2 (0.6) | 39 | 2.6 (1.7) | -1.67 | 87 | 0.09 | 55 | 2.1 (0.9) | 44 | 2.4 (1.1) | -1.20 | 97 | 0.23 |
| Use of emotional support | 143 | 5.1 (2.1) | 142 | 4.8 (2.1) | 1.18 | 283 | 0.23 | 50 | 5.3 (1.8) | 39 | 5.2 (2.1) | 0.22 | 87 | 0.81 | 57 | 5.4 (2.0) | 44 | 5.6 (2.1) | -0.48 | 99 | 0.63 |
| Use of instrumental support | 145 | 5.5 (2.2) | 142 | 5.1 (2.1) | 1.58 | 285 | 0.11 | 50 | 6.3 (1.8) | 39 | 5.4 (2.2) | 2.09 | 87 | 0.03 | 57 | 5.7 (2.1) | 44 | 6.1 (2.1) | -0.93 | 99 | 0.35 |
| Behavioural disengagement | 138 | 4.1 (1.6) | 137 | 3.6 (1.7) | 2.25 | 273 | 0.02 | 48 | 4.2(1.5) | 39 | 4.4 (1.6) | -0.72 | 85 | 0.47 | 57 | 4.0 (1.5) | 42 | 4.2 (1.8) | -0.65 | 97 | 0.51 |
| Venting | 139 | 4.5 (1.8) | 139 | 4 (1.9) | 1.86 | 276 | 0.06 | 50 | 4.5 (1.7) | 38 | 4.6 (1.9) | -.180 | 86 | 0.85 | 56 | 4.6 (1.8) | 43 | 5.0 (1.8) | -1.15 | 97 | 0.25 |
| Positive reframing | 141 | 4.5 (2.1) | 143 | 4.4 (2.2) | 0.55 | 282 | 0.57 | 50 | 5.1 (1.9) | 37 | 4.7 (2.0) | 1.07 | 85 | 0.28 | 57 | 5.1 (1.7) | 44 | 5.7 (1.8) | -1.60 | 99 | 0.11 |
| Planning | 143 | 5.6 (2) | 142 | 5.3 (2.1) | 1.10 | 283 | 0.27 | 50 | 6.4 (1.6) | 37 | 6.2 (1.8) | 0.42 | 85 | 0.67 | 57 | 6.3 (1.7) | 43 | 6.6(1.4) | -0.93 | 98 | 5 |
| Humour | 141 | 2.8 (1.3) | 141 | 2.6 (1.4) | 1.34 | 280 | 0.18 | 50 | 3.2 (1.5) | 39 | 3.1 (1.8) | 0.32 | 87 | .74 | 57 | 2.9 (1.3) | 44 | 3.1(1.5) | -0.70 | 99 | 0.48 |
| Acceptance | 141 | 4.8 (2.1) | 141 | 4.2 (2.1) | 2.33 | 280 | 0.02 | 50 | 4.9 (1.5) | 38 | 5.4 (1.8) | 1.30 | 86 | 0.19 | 56 | 4.9 (1.6) | 43 | 5.6 (1.8) | -2.04 | 97 | 0.04 |
| Religion | 142 | 4.8 (2) | 141 | 4.4 (2) | 1.53 | 281 | 0.12 | 50 | 4.7 (1.8) | 37 | 4.7 (2.0) | -0.08 | 85 | 0.93 | 57 | .7 (1.9) | 44 | 4.8 (2.0) | -0.32 | 99 | 0.74 |
| Self-blame | 138 | 4.7 (2) | 143 | 4.2 (2) | 2.03 | 279 | 0.04 | 50 | 4.2 (1.6) | 39 | 4.4 (1.9) | -0.55 | 87 | 0.58 | 57 | 4.2 (1.6) | 44 | 4.2 (2.1) | 0.04 | 99 | 0.96 |

use of adaptive coping methods in both the intervention and control groups, while others did not experience any changes in coping and were still struggling with problems. Most participants stated that they used multiple coping methods together when dealing with current life problems after the self-poisoning incident.

The findings from the interviews revealed positive changes in the relationship patterns of participants in both BPI and TAU groups following the self-poisoning incident. Specifically, participants were more likely to talk to others and seek advice or social support from others when dealing with problems. This was mostly identified among the younger age group between their parents and friends, but also for some older women and mainly reported by the younger participants and appeared as a stronger theme in the intervention group.

> "(Previously) I didn't tell my parents, [siblings], cousins, or anyone close to me. Now I feel that if I had spoken with them, they could have provided a solution. I wouldn't hesitate if there were any more problems. I'd tell Mother, and she'll show me a way to move on from the problem. I'll tell an adult; get any solution they offer and try to stay free of the burden in my mind." (P1-intervention, T2)

Some participants reported that they had adapted to their problems in a different way – they felt they had no choice but to live with the problem. This strategy of 'enduring a problem by thinking about the children' was reported by married women as a way of dealing with family problems. One married woman stated:

*"That's his (husband's) behaviour. I try to adapt myself to their (husband's) lifestyle because it's the way to continue with family life. That problem can never be solved. So, I just made up my mind and tolerated it." (P2-intervention group, T3).*

Expressing negative emotions was another theme that was described by both groups. In some instances, this was mal-adaptive, with some participants saying that they still behaved aggressively and with feelings of anger when they experienced problems, just as they did before the self-poisoning event. One woman who was in treatment for a mental illness described her behaviour of harming herself when she had problems with her husband or parents:

*"I worry too much. Keep hitting my head impulsively when alone. I don't know what I am doing." (P9- intervention group, T2)*

Utilising different activities was another coping strategy which was mainly reported by the participants from the BPI group. This method is similar to 'the self-distraction' coping strategy in the Brief COPE, which was significantly identified as being useful by participants in the intervention group, at six-month follow-up. Engaging in entertaining activities such as listening to songs and watching TV or movies was found to be more common among younger people. Older participants identified different kinds of distraction strategies. For example, gardening was highlighted by middle-aged married women as a way of forgetting about issues.

**Mechanisms of change.** The Brief COPE and the results from the interviews suggest that several changes in the use of coping strategies had occurred among many of the participants in both the BPI and TAU groups after the self-poisoning incident. Thus, it is worth looking at what mechanisms led to the changes and how the self-poisoning incidents influenced such changes.

A majority of participants experienced positive changes in their relationships with their close associates after the DSP incident. Many participants identified that these closer relationships provided a necessary foundation for changes in their coping skills, and nominated family and/or friends as important factors in the change process. Critically important was a perceived change in the quality of relationships: a friendlier mother, friends who listen more now, and help from parents that was not expected or experienced previously.

Several participants described that the self-poisoning incident itself and the time spent in the hospital for treatment led them to rethink their lives and better face problems. Sharing experiences with other self-poisoning patients who were in the hospital at the same time appears to have made an impact on participants and helped them realise that having problems is common for everyone. They shifted from thinking they were the only ones with such problems to realising that others have these issues too. One woman described:

"When in the hospital, a number of patients were around me. We talked to each other. Then I knew there were problems even worse than mine. I have one problem, but others have problems even with their child, husband and mother-in-law. This made me feel better" (P23- intervention group, T2)

Participants, especially the younger participants (age 16-25) in the intervention group, described that what they learned from the PBI intervention led them to change their coping skills positively. Another striking finding was that almost every participant from the intervention group remembered the calendar and the message they wrote at the end of the counselling session. When talking about the efficacy of the take-home message (Fig 3), one girl stated:

"…Sometimes I look at it. I am the one who wrote that message on the calendar. After I wrote my message on the calendar, the nurses who delivered the counselling drew a face with a lovely smile. I feel happy when I see it. I just see it, even while on a bus…The calendar is always useful for me when I am feeling sad or stressed." (P7, T3)

Not all participants found the intervention useful. In follow-up interviews, some participants stated that they could not remember the content of the intervention session they had received at the hospital, although they could remember that

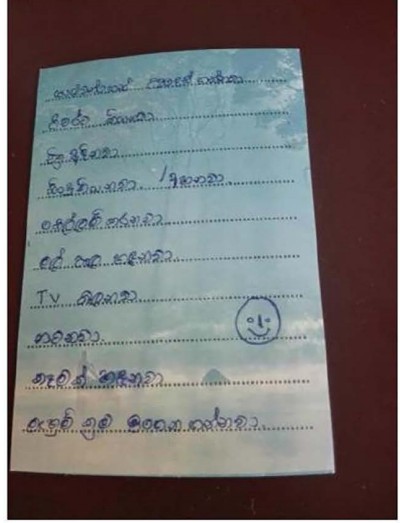 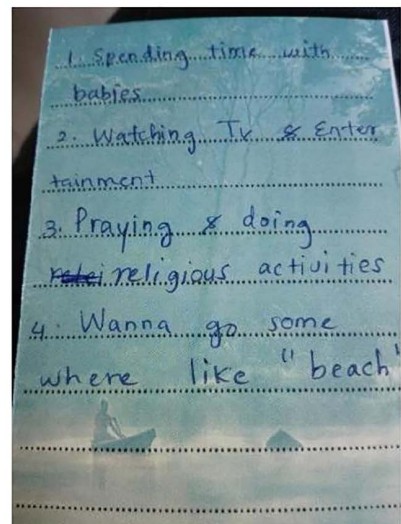

**Fig 3. The pocket calendar given to the participants.** This image shows the pocket calendar (with some handwritten messages) that was given to the participant to write down a brief take-home message at the end of the counselling session.

they had participated in it. This was particularly evident for a few older participants (60 > years of age) and two young girls who were receiving treatment for depression.

## The views and experiences of participants towards the brief counselling intervention

Participants in the intervention group of this study shared their views and attitudes toward the brief psychological intervention delivered by the trained designated nurse. The majority had positive attitudes towards the intervention, as described below. A feeling of relief was the main response that most of the participants described after receiving the BPI. The findings showed that most of the participants wanted to talk to a person about the difficulties they faced. An eighteen-year-old girl described her experiences with the intervention:

> *"So, at that time, talking to me…that gave me huge relief. The reason was that she listened to everything I told her. To whatever I said. That was a relief… giving a chance to tell the things in my heart and listening." (P 21, T3)*

Most were happy to have the opportunity to openly discuss their problems with the nurse while participating in the counselling session. This was particularly prominent among women who had faced problems with their married life. They reported that they could reveal their personal life issues to the nurse.

Several suggestions were given to further improve the intervention. Participants suggested involving their significant others in a BPI session in the future. More specifically, they believed it would be vital to counsel the person who triggered the self-poisoning act or the patient's close associates. Several married women discussed the importance of providing psychological or educational sessions to their husbands as a part of the intervention. Some highlighted the importance of others (family members or close associates) being aware of appropriate ways of dealing with a person who has self-harmed.

## Discussion

### Efficacy of the brief psychological intervention

In this study, we explored the impact of a brief psychological intervention for people who have engaged in deliberate self-poisoning, with regard to psychiatric morbidity. We found no significant changes in rates of depression, alcohol use

disorder, or suicidal ideation among the intervention and TAU, while the brief counselling intervention was efficacious in mitigating anxiety and enhancing the use of positive coping strategies at the six-month follow-up.

The low rate of psychiatric morbidity at baseline in our study, which is in keeping with findings from other studies on self-harm, might be a reason that the brief-counselling intervention did not impact significantly depression rates pre- and post-intervention [36–38]. This indicates the need for 'non-psychiatric' support for people at risk of self-harm, particularly younger people who are more likely to be responding with acute distress or acute anger in the context of stress, rather than psychiatric illness. In our study, older (>30 years) individuals screened positive for depression, similar to previous local findings [20]. Therefore, while older people with psychiatric morbidity are likely to need more specific psychiatric support, the larger proportion of younger people (16–25 years) without psychiatric morbidity may benefit from a different approach for support.

Furthermore, among the participants who screened positive for depression at baseline, only 23% and 27% participated in the six-month and one-year follow-ups, which may also have affected findings. Similar problems in recruiting and retaining depressed people into randomized controlled trials (RCTs) have been reported in other studies [39].

The original C-MAP study from Pakistan reported that C-MAP was associated with a significant reduction in suicidal ideation, hopelessness and symptoms of depression [8]. The original C-MAP study comprised six sessions within 3 months [8], whereas our study intervention was limited to one session, for pragmatic reasons (such as staff time limitations and the lower likelihood of participants attending after discharge). This may have impacted the different findings between our study and the Pakistan study.

In this study, the group that received the BPI had lower levels of anxiety and changes in some coping strategies at six months, compared to the TAU group, although this was not seen at one year. In terms of changes in coping skills, participants in the BPI group used seeking social support and self-distraction more often at the six-month follow-up, compared to the TAU group. These changes were not observed at the one-year follow-up. The impact of the intervention on anxiety and coping strategies appeared short-lasting, being seen at six months follow-up, but not at one year. This suggests the need for intermittent, ongoing psychological support for those at risk. Previous studies on the efficacy of brief psychological interventions for individuals presenting with self-harm and suicidal behaviour also emphasized the requirement for more rigorous and long-term follow-up, as most of these approaches prepare people with new skills that may take time for them to master and use in dealing with future stresses [11,40,41].

The self-written message, which the participants carried away with them, played an important role in reminding the person 'how to respond' in more adaptive ways – the fact that the message was 'self-generated' seemed to help participants relate to it more strongly. The younger participants, in particular, highlighted that the brief take-home message was useful at follow-up. Previous studies have evaluated that postcard or letter-writing interventions have beneficial effects on suicidal behaviours in a range of populations and countries [42–44]. The pocket calendar used in this study may have worked in a similar way because participants felt they were empowered and guided by the message in dealing with future stresses. Our study differed from the previous postcard intervention in that the message was generated by the study participants themselves, not by a healthcare provider. Of note is the fact that several participants felt the message was important because it was 'what they had written' – this personalized aspect appeared to be important in helping them keep the message and refer to it later.

This 'self-generated message' concept is a very simple intervention that could be used for future work that focuses on similar, cost-effective brief interventions to help people deal with distress or stressors; this could be especially effective in low-resource settings. This type of brief intervention could be implemented as a novel school-based, youth clubs or vocational training centres, community-based, or online-based preventive strategy. It could specifically be useful for helping young people develop coping skills and deal with emotional distress. The potential of delivering such intervention with the involvement of local stakeholders and through the Medical Officer of Health (MOH) Office should be further explored.

The results also showed that during the six-month and one-year follow, there had been a change in their coping strategies in *both* the intervention and control groups. This suggests that some of the changes in coping styles were not only

linked to the BPI but rather may have been driven by alternative factors, which are worth exploring further. Understanding such socio-cultural factors might be worthwhile in order to develop interventions at the primary level.

**Feasibility and acceptability of the brief psychological intervention**

The results of our study give the sense that BPI was valued by the majority because they felt someone listened to them, and they felt relieved talking about their problems. In particular, this response was mainly found among the young group (16–20 years age), who represent the major proportion of the study. This might be because of a sense of 'distance' from parents, family and friends [45,46]. Being there, being available to talk, and providing a sense of support was found to be helpful to young people who self-harm [47]. Similarly, our study also found that the main expectation of the participants was for the nurse to talk with them and to lend a supportive ear rather than find solutions to their problems.

Somewhat older women, who were married, also expressed relief in talking about their problems with the study nurse, especially when they were facing conflict with their husbands or a domestic violence situation. This may have given them relief. Concerns about social criticisms, feelings of shame and stigma may also have made it conducive to talk to the nurse in confidence. Family beliefs can also make it difficult for these women to specifically disclose and respond to domestic violence [48]. These factors may have influenced women to feel safe talking to the nurse.

While many participants were positive about the intervention, not all reported it as being useful or remembered it. Many people who mentioned that they could not remember the intervention were older (>55 years of age); some of them had with psychiatric illness. The psychiatric illness and some cognitive effects of the poisoning may have also influenced this finding.

The delivery of the intervention was challenging for the ward nurses in general at the start, due to the time and other commitments. Their contributions also varied with nurses' aptitude and interest in providing psychological care. During the first part of this study, it was observed that a few nurses who had personal skills and interest in this new role carried out the counselling role even with the obstacles and ward commitments. Thus, the aptitude and interest of the nurse concerned influenced their engagement in the counselling process. Accordingly, the original premise of requiring all nurses in a busy medical ward to provide counselling to patients after self-harm was not feasible or practical, although the nurses were accepting and even enthusiastic about the idea. The more feasible option that this study ended up adopting was to have a dedicated, trained nurse, who had an interest in counselling, deliver the counselling intervention.

At an administrative level, having a 'new' or separate nurse for counselling in the healthcare setting is likely to be a challenge. Within the current health setup in Sri Lanka, there are already nurses who play a role in public health education and health promotion -perhaps their role could be broadened to include this type of counselling as well. A different sort of challenge might be the attitudes of healthcare staff (at all levels) – they may not deem this kind of intervention to be necessary, or may 'blame' people who self-harm as being 'less deserving' of counselling support, compared to say a lactation nurse who give advice and support to a woman with a new-born child who is struggling to breastfeed. Thus, there is also a need to increase awareness among healthcare staff about the challenges associated with self-harm and how people could be provided with support to minimize this phenomenon.

**Strengths and limitations of the study**

This study used an RCT with an embedded qualitative research design for the purpose of confirmation, corroboration and cross-validation within a single study. DSH is a very complex phenomenon in need of further understanding from multiple perspectives. With this in mind, the mixed-method design enabled us to explore and integrate different types of data to determine whether the intervention was effective and feasible and for whom it was more useful though participants were not asked to directly review or provide feedback on the findings.

In terms of using the data collection tools used for the study, there were no locally validated versions of this scale available of the GAD-7 and Brief COPE available at the time of the study, but they were translated into Sinhala and back-translated and checked for accuracy before use.

The original C-MAP study from Pakistan was a brief problem-focused therapy intervention comprising six sessions delivered within three months, whereas the BPI was delivered as a single session, which may have impacted the efficacy.

There was some difference in a few aspects of coping skills among the participants in the BPI and TAU groups at baseline as per the Brief COPE, which is a limitation - but this impact was minimized with randomisation of participants into the case and control groups and additionally, stratified analysis showed no significant difference between people in the BPI and the TAU group.

Another limitation of this study was that the attrition rate was high in the follow-up assessments at six months and one year, respectively. This might have influenced the outcomes associated with depression, repeated episodes, and suicidal thoughts. Additionally, these low numbers limited the ability to further analyse the data. The required sample size was calculated to allow for a 50% drop-out rate, but the attrition rate in this study was very high. High attrition was may have been mainly because of the young and highly mobile nature of the sample, frequent changes in contact information, and the stigma associated with being recontacted about a self-harm episode; factors that have similarly affected follow-up in other Sri Lankan self-harm studies. Nonetheless, every effort was made to encourage participants to attend follow-up visits.

In addition, due to the high attrition at follow-up, some originally continuous outcome measures (PHQ-9, GAD-7, AUDIT, and Brief COPE) were re-categorised to maintain adequate numbers for analysis. This may have reduced the ability to detect differences between groups, and where differences were observed, the magnitude of the effects may have been underestimated. Future studies with larger sample sizes may allow for the analysis of these outcomes as continuous variables, providing more detailed insights.

## Conclusions

The BPI used in this study is useful in reducing anxiety in the first six months following an attempt at self-harm, and showed a short-term impact in enhancing coping skills after self-harm, among young people (age 16–25 years). People who have self-harmed in the context of depression or high suicide intent require a more specific, probably psychiatric support. It is notable that most people who self-harm in Sri Lanka are young people, and many have no psychiatric morbidity. For this majority, this type of brief counselling intervention appears useful to provide emotional support, reduce anxiety and enhance coping. An unexpected highlight was the calendar, with its personalized take-home message – this stood out particularly as being helpful, and is akin to the findings of the previous postcard interventions on suicidal behaviours [42–44]. It is worth further exploring the efficacy and feasibility of this kind of counselling with a designated nurse or another health professional in different healthcare settings, including rural and primary care settings, with the involvement of relevant stakeholders.

## Supporting information

**S1 Appendix. Summary of the most frequently reported responses from thematic analysis.**
(DOCX)

**S1 Checklist. CONSORT 2010 Checklist.**
(PDF)

**S2 Checklist. COREQ Checklist.**
(PDF)

## Acknowledgments

The authors would like to acknowledge the support given by the South Asian Clinical Toxicology Research Collaboration (SACTRC), Faculty of Medicine, University of Peradeniya, Sri Lanka, for implementing the study, and in particular Dr. Fahim Cader, Seyed Shahmy, Dilani Pinnaduwa, Indunil Abeyratne, and Sujani Ekanayake, Imalsha Wickramasuriya, for

their support in setting up the study. The authors would also like to give thanks to the staff at the Teaching Hospital Peradeniya for accommodating this research.

## Author contributions

**Conceptualization:** Lakmini De Silva, Judi Kidger, Sampath Thennakoon, Andrew Dawson, Indika Gawarammana, Thilini Rajapakse.

**Data curation:** Lakmini De Silva, Judi Kidger, Sampath Thennakoon, Andrew Dawson, Indika Gawarammana, Thilini Rajapakse.

**Formal analysis:** Lakmini De Silva.

**Funding acquisition:** Sampath Thennakoon, Andrew Dawson, Thilini Rajapakse.

**Investigation:** Lakmini De Silva, Thilini Rajapakse.

**Methodology:** Lakmini De Silva, Judi Kidger, Sampath Thennakoon, Andrew Dawson, Indika Gawarammana, Thilini Rajapakse.

**Project administration:** Lakmini De Silva, Thilini Rajapakse.

**Supervision:** Judi Kidger, Sampath Thennakoon, Andrew Dawson, Indika Gawarammana, Thilini Rajapakse.

**Validation:** Lakmini De Silva, Judi Kidger, Sampath Thennakoon, Thilini Rajapakse.

**Visualization:** Lakmini De Silva.

**Writing – original draft:** Lakmini De Silva.

**Writing – review & editing:** Lakmini De Silva, Judi Kidger, Sampath Thennakoon, Andrew Dawson, Indika Gawarammana, Thilini Rajapakse.

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
