## [Decision Letter · Decision Letter 0]

9 Sep 2025

PMEN-D-25-00315

Brief psychological intervention for the prevention of deliberate self-poisoning:  A  randomized controlled trial from Sri Lanka

PLOS Mental Health

Dear Dr. De Silva,

Thank you for submitting your manuscript to PLOS Mental Health. After careful consideration, we feel that it has merit but does not fully meet PLOS Mental Health’s publication criteria as it currently stands. Therefore, we invite you to submit a revised version of the manuscript that addresses the points raised during the review process.

The manuscript has been evaluated by two reviewers, and their comments are available below.

The reviewers have raised a number of concerns that need attention. They have questions about the analysis of data, with a need for statistical explanations.

Could you please revise the manuscript to carefully address the concerns raised?

We look forward to receiving your revised manuscript.

Kind regards,

Jenna Scaramanga

Staff Editor

PLOS Mental Health

Journal Requirements:

Additional Editor Comments (if provided):

Reviewers' comments:

Reviewer's Responses to Questions

**Comments to the Author**

1. Does this manuscript meet PLOS Mental Health’s publication criteria?

Reviewer #1: Yes

Reviewer #2: Partly

2. Has the statistical analysis been performed appropriately and rigorously?

Reviewer #1: Yes

Reviewer #2: No

3. Have the authors made all data underlying the findings in their manuscript fully available (please refer to the Data Availability Statement at the start of the manuscript PDF file)?

Reviewer #1: Yes

Reviewer #2: Yes

4. Is the manuscript presented in an intelligible fashion and written in standard English?

Reviewer #1: Yes

Reviewer #2: Yes

Reviewer #1: Interesting paper. Relatively large sample size. The effect is modest but the intervention is very cheap. The study may inspire to similar interventions in other countries. Many were lost during follow-up. How come?

Reviewer #2: This was a mixed method design study which the investigators labeled as a a pilot randomized controlled trial (RCT) with embedded qualitative methods. The quantitative requirement of sample size was based on the continuous t-test format. However, the study analysis appeared to be based mostly on categorization of some of the major endpoints, depression, AUDIT and anxiety.

1. The protocol (Statistical Consideration) sample size is based on proportions. Please reconcile this.

2. What was the rationale for categorization of some of the endpoints?

The analysis could have been more sophisticated using the continuous endpoints. There were two interventions and three time points. There is a loss of information with categorization as the authors should know.

3. Why not a repeated measures design with covariate adjustment , if needed?

As noted above, there seems to be somewhat of a disconnect between the protocol in the supplement and the text in the paper.

4. The efficacy review in the protocol (11.3.2) mentions the relative risk (RR). Where is that?

The authors mention attrition as a limitation, but talk about imputation in the protocol?

5. What was the intended imputation? Was it explored?

There may be some typos in the paper. Please check for those.

6. For example, Table 4, six month GAD-7 for BPI the percent (88.4 + 7.6) does not add to 100, unless this reviewer missed something.

The qualitative information is somewhat informative. Some tabulations of the more common suggestion or responses noted might be helpful.

There was a lot of effort put into this important project by the investigators. The clarifications requested above would be helpful?

**Do you want your identity to be public for this peer review?** For information about this choice, including consent withdrawal, please see our Privacy Policy

Reviewer #1: No

Reviewer #2: No

---

## [Decision Letter · Decision Letter 1]

4 Feb 2026

Brief psychological intervention for the prevention of deliberate self-poisoning:  A randomized controlled trial from Sri Lanka

PMEN-D-25-00315R1

Dear Dr De Silva,

We are pleased to inform you that your manuscript 'Brief psychological intervention for the prevention of deliberate self-poisoning:  A randomized controlled trial from Sri Lanka' has been provisionally accepted for publication in PLOS Mental Health.

Best regards,

Johanna Pruller, Ph.D.

PLOS Staff Editor

PLOS Mental Health

Reviewer Comments (if any, and for reference):

Reviewer's Responses to Questions

**Comments to the Author**

Reviewer #2: All comments have been addressed

publication criteria?

Reviewer #2: (No Response)

3. Has the statistical analysis been performed appropriately and rigorously?

Reviewer #2: (No Response)

4. Have the authors made all data underlying the findings in their manuscript fully available (please refer to the Data Availability Statement at the start of the manuscript PDF file)?

Reviewer #2: (No Response)

5. Is the manuscript presented in an intelligible fashion and written in standard English?

Reviewer #2: (No Response)

Reviewer #2: (No Response)

**Do you want your identity to be public for this peer review?** For information about this choice, including consent withdrawal, please see our Privacy Policy

Reviewer #2: No
